# Qualitative interview study of parents' perspectives, concerns and experiences of the management of lower respiratory tract infections in children in primary care

Amy Halls,[1] Catherine van't Hoff,[2] Paul Little,[1] Theo Verheij,[3] Geraldine M Leydon[1]

[1]Faculty of Medicine, Academic Unit of Primary Care and Population Sciences, Aldermoor Health Centre, University of Southampton, Southampton, UK
[2]Education Centre, Royal Bournemouth Hospital, Bournemouth, UK
[3]Department of General Practice, University Medical Center Utrecht, Julius Center for Health Sciences and Primary Care, Utrecht, The Netherlands

**Correspondence to**
Professor Geraldine M Leydon;
gerry@soton.ac.uk

## ABSTRACT

**Objective** To explore parents' perspectives, concerns and experiences of the management of lower respiratory tract infections (LRTIs) in children in primary care.

**Design** Qualitative semistructured interview study.

**Setting** UK primary care.

**Participants** 23 parents of children aged 6 months to 10 years presenting with LRTI in primary care.

**Method** Thematic analysis of semistructured interviews (either in person or by telephone) conducted with parents to explore their experiences and views on their children being prescribed antibiotics for LRTI.

**Results** Four major themes were identified and these are perspectives on: (1) infection, (2) antibiotic use, (3) the general practitioner (GP) appointment and (4) decision making around prescribing. Symptomatic relief was a key concern: the most troublesome symptoms were cough, breathing difficulty, fever and malaise. Many parents were reluctant to use self-care medication, tended to support antibiotic use and believed they are effective for symptoms, illness duration and for preventing complications. However, parental expectations varied from a desire for reassurance and advice to an explicit preference for an antibiotic prescription. These preferences were shaped by: (1) the age of the child, with younger children perceived as more vulnerable because of their greater difficulty in communicating, and concerns about rapid deterioration; (2) the perceived severity of the illness; and (3) disruption to daily routine. When there was disagreement with the GP, parents described feeling dismissed, and they were critical of inconsistent prescribing when they reconsult. When agreement between the parent and the doctor featured, parents described a feeling of relief and legitimation for consulting, feeling reassured that the illness did indeed warrant a doctor's attention.

**Conclusion** Symptomatic relief is a major concern for parents. Careful exploration of expectations, and eliciting worries about key symptoms and impact on daily life will be needed to help parents understand when a no antibiotic recommendation or delayed antibiotic recommendation is made.

## Strengths and limitations of this study

► Provides a detailed view of key experiences and understandings of a sample of parents of children presenting with symptoms of lower respiratory tract infection in southern England.
► Only 2 fathers included (compared with 23 mothers) so the views of fathers were not fully captured.
► Semistructured interviews were the optimal data collection method given the aims of the research and analytical saturation was reached.

## BACKGROUND

The majority of respiratory tract infections (RTIs) are viral[1,2] and will resolve spontaneously with analgesia and rest. Infections and treatment with antibiotics are important health concerns.[3] A third of children presenting to a general practitioner (GP) with febrile illness receive an antibiotic prescription.[4] NICE (National Institute for Health and Care Excellence) guidance recommends patients be given no antibiotic prescription or a delayed prescription (to be fulfilled if symptoms worsen),[5] as well as advice about expected time course of infection and how to manage symptoms. Patients' education and general understanding of the medications they are prescribed affect compliance[6] and use of antibiotics leads to greater risk of development of antibiotic-resistant bacteria.[6,7] A US survey study of parents of children under 4 years suggested that parents often believed antibiotics were indicated for cough and upper RTIs,[8] and yet most parents in the USA and UK are aware of antibiotic resistance and its complications[8,9] or express concern about the overuse of antibiotics. More than half of adults may expect an antibiotic prescription when attending with symptoms of RTI.[6]

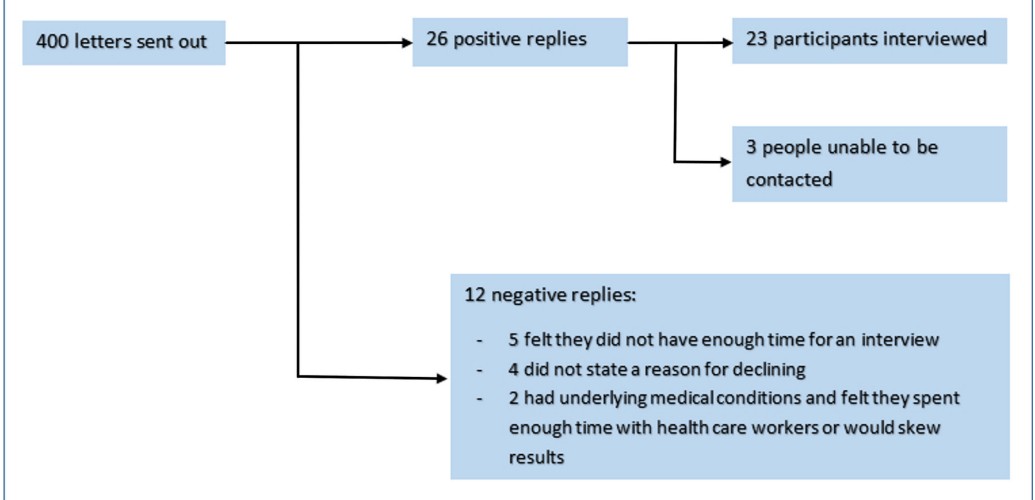

**Figure 1** Recruitment process.

Uncomplicated acute lower respiratory tract infection (LRTI) is the most common acute infection,[10] and is likely to have the longest and highest symptom burden of all acute RTIs.[11] Previous work has assessed parents' understanding of the implications and indications for an antibiotic prescription,[12] with parents believing that antibiotics were for more severe illnesses, supported by GP's explanations. This study provides further insight into the tension between GPs and parents in their decision making, and explores key drivers of parental expectation and beliefs about antibiotics.

We report a qualitative study of the key concerns of parents about symptoms of their children presenting with acute LRTI in primary care and their perceptions about antibiotics and the consultation.

## METHODS
### Participants and procedure
Six research-active practices in the South West of England were identified through the Primary Care Research Network and a poster presentation by CvH at the Wessex Research Sites Initiative Conference. Practices searched their databases for children aged 6 months to 12 years who had presented to primary care with symptoms of LRTI between January 2013 and March 2015. The recruitment process is shown in figure 1. A purposive sample was originally intended: however, due to initially slow recruitment, a convenience sampling strategy was used. While this sampling approach can be limiting, we recruited a good range of mothers in terms of age and occupation. Both urban and rural practices were invited to participate; however, a deprivation scale was used to illustrate variation, as shown in table 1. Parents were interviewed regardless of the method of GP consultation or whether antibiotics were prescribed.

### Interviews
Two female interviewers (CvH and AH) conducted face-to-face interviews in the participant's home (n=20) and telephone interviews (n=3), each lasting approximately 30–60 min, with an average duration of 42 min. Both interviewers were aware of the NICE guidance regarding antibiotic prescribing for RTIs, and made field notes during interviews. All interviews were audio-recorded and transcribed verbatim. Qualitative interviews provided the best method for gathering insights into parents' views about their children's experiences of RTI and its management in primary care. The interview guide (online supplementary appendix 1) included key topic areas, not all of which are covered in this paper. Interviews explored parent's views on whether they

| Practice | Total letters sent | Positive replies | Negative replies | Total replies | Number consented | Deprivation score* |
|---|---|---|---|---|---|---|
| 1 | 33 | 1 | 2 | 3 | 1 | 13.5 |
| 2 | 106 | 3 | 1 | 3 | 2 | 19 |
| 3 | 44 | 3 | 0 | 3 | 2 | 22.8 |
| 4 | 85 | 9 | 3 | 12 | 9 | 7.1 |
| 5 | 12 | 2 | 0 | 2 | 3 | 13.6 |
| 6 | 120 | 7 | 6 | 12 | 6 | 13.3 |

**Table 1** Practice characteristics and recruitment

*National General Practice Profiles*, https://fingertips.phe.org.uk/profile/general-practice/data. Higher score indicates a higher level of deprivation.

would be willing to be randomised in a future study. Its semistructured design gave flexibility to explore new areas if they arose.

## Analysis

Inductive thematic analysis[13] was conducted on all transcripts to gain an understanding of the perspectives, attitudes and concerns of parents regarding the managing of LRTI in children in primary care. CvH and AH achieved familiarisation through repeated reading (and listening) of the transcripts. Patterns and themes in the data were identified by CvH and AH and labelled with codes: these code labels referred to the operationalisation of the theme content. A label and full description were provided for each theme. These codes and definitions were refined during a continuous process led by GML, CvH and AH: this involved themes being linked, grouped, moved, relabelled, added and removed as appropriate in order to produce a set of themes, subthemes and a coding manual, reviewed and confirmed by the full research team. The coding manual thoroughly explained all the data.

## Participants

In total, 23 participants took part in this study. In all interviews, the mother of the child was interviewed, and fathers were present in two interviews; however, anyone who fulfilled a caring role could have participated. The age of the mothers[i] ranged from 21 to 47 (median age 34) and the age of the children ranged from 6 months to 10 years (median age 1 year 9 months) (see table 2 for participant characteristics).

**Table 2** Participant characteristics

| Participant | Parent(s) interviewed | Age of mother | Age of child when ill |
|---|---|---|---|
| 1 | Mother and father | 21 | 6 months |
| 2 | Mother | 33 | 3 years |
| 3 | Mother | 38 | 10 years |
| 4 | Mother | 36 | 1 year |
| 5 | Mother | 30 | 1 year |
| 6 | Mother | 39 | 1 year |
| 7 | Mother | 31 | 2 years |
| 8 | Mother | 41 | 6 months |
| 9 | Mother | 33 | 1 year |
| 10 | Mother | 34 | 1 year |
| 11 | Mother | 20 | 2 years |
| 12 | Mother | 36 | 8 years |
| 13 | Mother | 28 | 9 years |
| 14 | Mother and father | 39 | 1 year |
| 15 | Mother | 47 | 1 year |
| 16 | Mother | 33 | 5–6 months |
| 17 | Mother | 39 | 2 years |
| 18 | Mother | 34 | 8 years |
| 19 | Mother | 28 | 1 year |
| 20 | Mother | 26 | 8 years |
| 21 | Mother | 42 | 8 years |
| 22 | Mother | 38 | 1 year |
| 23 | Mother | 46 | 8 years |

## FINDINGS

Thematic analysis identified four themes relating to factors which enhanced our understanding of parents' perspectives, concerns and experiences of the management of LRTI in children in primary care. Table 3 outlines the themes and subthemes and these are used to structure the findings.

The following sections describe each theme in turn. Numbers in parentheses refer to the theme, subtheme and relevant exemplary quotation provided in online supplementary appendix 2.

### Parental perspectives on infection

*Parental concerns:* Parents spoke in detail about their concerns when their child was ill. By far the strongest and most frequent concern centred on breathing difficulties and airway limitations (1.1.1). Other concerns included general concerns or unhappiness that their child was unwell (1.1.2). However, some mothers were less concerned and this was associated with factors such as the child not being their first child, or if the child was perceived to be receiving appropriate treatment (1.1.3). Parents reported worry about symptoms their child had, including the duration of

symptoms (1.3.4) or the qualitative difference in the child's behaviour (1.1.5).

Parents of younger children (2 years and under) generally had more concerns due to their child's young age. These concerns centred on three areas: increased vulnerability (1.1.6); difficulties in communicating (1.1.7); and the potential for rapid deterioration (1.1.8). Some parents also expressed worry about complications (1.1.9).

*Impact of child's illness:* Parents were directly asked about the impact of the child's illness on them and their family and readily gave lack of sleep and tiredness across the family as the primary impact (1.2.1). Parents of children with underlying health conditions spoke in more detail about the impact of their child's condition (1.2.2). Working mothers spoke of social implications such as needing additional time off work and the subsequent financial impact (1.2.3). These pressures were also reflected when mothers talked about the need for antibiotics to shorten illness duration (see theme 2.1).

*Symptoms:* All symptoms reported by parents are summarised in table 4, which shows that the most commonly reported symptom by the study participants was a cough, but breathing, fever and malaise were also very commonly reported as troublesome symptoms.

---

[i]The age of the fathers was not recorded.

**Table 3** Themes identified in analysis

| Themes | Subthemes |
|---|---|
| 1. Parental perspectives on infection | 1.1 Symptoms<br>1.2 Previously similar infections<br>1.3 Parental concerns<br>1.4 Impact of child's illness<br>1.5 Home management strategies and medicine usage |
| 2. Parental perspectives on antibiotic use for lower respiratory tract infection | 2.1 Benefits and efficacy<br>2.2 Parental concerns |
| 3. The GP appointment | 3.1 Hopes and expectations<br>3.2 Positive/negative experiences<br>3.3 Access to healthcare<br>3.4 Advice |
| 4. Decision making | 4.1 Agreement/disagreement for an AB prescription<br>4.2 Perspectives of GP's prescribing behaviour<br>4.3 Parental knowledge of LRTI and their management |

AB, antibiotic; GP, general practitioner; LRTI, lower respiratory tract infection.

*Previously similar infections:* During the course of the interview, parents recounted previous infections or the prolonged nature of the infection in question (1.2.1). Mothers of children with underlying health conditions

**Table 4** Reported symptoms

| Symptoms | Frequency |
|---|---|
| Cough | 18 |
| Breathing difficulties | 16 |
| Fever/temperature | 16 |
| Malaise* | 15 |
| Difficulty sleeping | 11 |
| Feeding problems/appetite | 10 |
| Wheeze | 10 |
| Runny nose | 9 |
| Fatigue/tiredness | 6 |
| Vomit | 6 |
| Sore throat/hoarse voice | 5 |
| Earache | 2 |
| Pain | 2 |
| Headache | 1 |
| Fainting | 1 |
| Heart racing | 1 |
| Rash | 1 |

*Malaise was used as a term to categorise anything parents said referring to a change in behaviours, character or appearing unwell (1.1.1).

**Table 5** Home management strategies tried in addition to Calpol/Nurofen

| Strategy | Interview mentioned in |
|---|---|
| Steam inhalation | 4,5,6,8,10,11,13,16 |
| Vaporisers/scents | 6,8,13,16 |
| Throat pastilles/cough syrup | 3,4,12,14,16,17 |
| Drinks | 3,4,8,12,19,21 |
| Cold strategies | 13,14,16 |
| Raise the bed | 8,16,17,19,23 |
| Vicks | 5,8,10,16, 21 |
| Improved diet | 5 |
| Other* | 2 |
| No additional home management | 1,7,9,15,20,22 |

*Participant 2's child took daily prophylactic antibiotics for a severe undiagnosed health condition: their home management strategy was to give a double dose, as recommended by the specialist.

often spoke of this in detail, but it is not central to the current study and so is not reported.

*Home management strategies and medicine usage:* Parents initially dealt with their child's infection in different ways. Parents reported trying home management strategies (see table 5) or complementary medicines before visiting the GP (1.5.1). Parents' views on over-the-counter medicines were varied (1.5.2) although many parents used Calpol and Nurofen[ii] to relieve symptoms, most commonly a temperature. Perhaps unsurprisingly many parents reported reluctance to use medications unless they felt it was strongly needed or recommended by a doctor (1.5.3).

## Parental perspectives on antibiotic use for lower respiratory tract infection

Parents spoke in detail about their perceptions of antibiotic use, their attitudes to prescribing and their concerns about antibiotics. Their views were varied but overall parents were happy with their experiences of antibiotic usage in their children.

*Benefits and efficacy:* Parents cited antibiotics as having three main benefits. First, shortening illness duration (2.1.1), which had the additional benefit of helping parents to return to work (2.1.2). Second, antibiotics were described as being able to prevent complications (2.1.3). Third, antibiotics were thought to provide symptomatic relief (2.1.4–2.1.5). Most parents expressed certainty that antibiotics were effective (2.1.6) and they often used their experience of their child recovering during past episodes to infer the effectiveness of antibiotics (2.1.7), although one parent did appreciate there were conditions in which antibiotics are more effective (2.1.8). In contrast, a minority of parents were aware

[ii]Analgesics widely available in the UK: Calpol contains paracetamol and Nurofen contains ibuprofen and codeine phosphate.

 Halls A, et al. BMJ Open 2017;7:e015701. doi:10.1136/bmjopen-2016-015701

that antibiotics are not always effective while some volunteered awareness of the lack of evidence for prescribing in children (2.1.9).

*Parental concerns:* Parental concerns about antibiotic use were predominantly about the composition of antibiotic medication (taste, texture, smell and use of additives). Many parents commented negatively on the taste and texture whereas views on additives such as colourings were more varied. Generally, parents had few concerns about adverse effects (2.2.1), especially when their child had not experienced any following previous use (2.2.2).

## The GP appointment

*Hopes and expectations:* Parent interviews indicated varied hopes and expectations of the outcome of their GP appointments. Many parents described a desire for symptomatic relief, reassurance and/or advice (3.1.1, 3.1.2) while some, importantly in the minority, attended the GP consultation with an explicit wish for an antibiotic prescription (3.1.3).

*Positive and negative experiences:* Many parents had positive views of the GP appointment; in particular when GPs were perceived to take time for a thorough assessment of the child (3.2.1) or with GPs who have built a history with the family (3.2.2). A physical examination has previously been shown to be a key expectation of parents.[9] However, some parents used the interviews as an opportunity to voice frustrations over 'bad' appointments; these centred around feeling rushed, appointments running late or parental concerns not being taken seriously. Differences in parental and GP opinions also contributed to dissatisfaction (discussed further under theme 4).

*Access to healthcare:* Parents reported irritation about difficulties in getting appointments. They also discussed a lack of continuity of care, which could mean that deterioration was not noticed by the GP and parents had to recount their story on multiple occasions. Many spoke of needing to use out-of-hours services, and this brought with it new concerns around queueing, and seeing an unfamiliar doctor in an unfamiliar environment. One parent, living rurally, spoke of concerns around rapid deterioration out of normal working hours. However, many parents spoke positively of the use of the NHS 111 service and their advice. Parents of children with underlying health conditions spoke of easy access to medical expertise (3.3.1).

*Advice:* Parents were asked to discuss advice they were given at the index appointment. They primarily volunteered advice on home management approaches they had already tried prior to the GP consultation (3.4.1). Many parents did not recall being given specific advice about the natural history of the illness. Of those interviewed, some were asked to reconsult if there was no improvement and some recalled receiving advice on how long symptoms would last (3.4.2).

## Decision making

A shared decision-making process between parent and GP, or an explanation of why antibiotics were not necessary, usually led to a more acceptable consultation when parents did not explicitly express a desire for antibiotics.

*GP–parent agreement and disagreement about the need for an antibiotic prescription:* When there was agreement about the need for a prescription between GP and parents, many parents described a feeling of relief. A prescription seemed to give parents the feeling of illness legitimation[1] (4.1.1, 4.1.2). In addition to this, parents reported feeling empowered by a prescription because it signalled a positive action taken to help their child (4.1.3). Parents also described that they were pleased because their child would now have symptomatic relief (4.1.4, 4.1.5). However, parents of children with frequent chest infections or underlying health conditions did not seem to have strong opinions, viewing an antibiotic prescription as a necessary requirement (4.1.6).

When parents felt there was disagreement between them and their doctor, feelings of frustration were described. Frustration that doctors did not agree with their perception of the severity of infection (4.1.7), as there is a perceived link between the need for antibiotics only in severe infections,[12] or frustration that doctors did not think antibiotics were indicated (4.1.8). Parents felt frustrated and upset at the lack of symptomatic relief or that they would need to reconsult (4.1.9, 4.1.10).

Parents spoke about feeling dismissed by their GP, especially if the infection was thought to be viral (4.1.11), as seen in previous studies.[12] Parents reported uncertainty when they disagreed with the doctor's decision, but this narrative was balanced by belief in ultimately trusting the doctor's judgement and their professional expertise (4.1.12–4.1.14).

Some parents described discomfort and the feeling they could not question the GP's judgement (4.1.15). The majority of parents trusted doctors' expertise but many parents recounted occasions when they had not been prescribed antibiotics initially and then had reconsulted shortly after and been given a prescription. By having to reconsult, parents felt doctors were not consistent, and that their persistence or pestering would result in a prescription. This often seemed to be closely related to dissatisfaction with the GP appointment (4.1.16). Interestingly, it is noted that GPs can feel increased pressure and are more likely to prescribe if a parent has reconsulted.[14]

*Perspective of GP's prescribing behaviour:* Many parents had an awareness of overprescribing antibiotics and its consequences and some commented on doctors clearly trying to limit prescribing (4.2.1). Many parents understood antibiotic resistance to be an idea that overuse would result in your body becoming immune to the antibiotic; an idea which is echoed in the existing literature (4.2.2).[12 15]

Parental concerns about overuse were reflected in their understanding of antibiotic resistance and superbugs. For example, parents with more accurate scientific

knowledge often had a clear appreciation of the consequences of resistance (4.2.3). However, some parents were quite clear that they had very limited understanding of the concept of antibiotic resistance (4.2.4).

Parents offered both positive and negative opinions on GP's prescribing behaviours (4.2.5). Negative opinions of prescribing behaviour were often from word of mouth. For example, a few parents spoke about factors they believed to influence prescribing (4.2.6) such as believing doctors earn more if they prescribe fewer antibiotics (4.2.7).

*Parental understanding of LRTI and their management:* Parents had a degree of prior knowledge regarding a suggested diagnosis, when to present and if they felt antibiotics were indicated. Many parents spoke of knowing their child best (4.3.1), although this contrasted with previous comments, where parents described difficulties in communicating with young children. Parents appeared to have had different 'tipping points' with regard to help seeking, such as when to present. Some parents tended to have a particular amount of time they were happy waiting before they consulted to help ensure they received antibiotics (ie, illness duration would be such that a GP would be more likely to think antibiotics are required) (4.3.2). Others waited a specific number of days (4.3.3) or used a lack of improvement as a reason to consult (4.3.4). Although many parents had clear triggers for consulting, they simultaneously reported concerns about wasting doctors' time (4.3.5), or appearing overanxious (4.3.6).

Parents often described having a good idea of the diagnosis before presenting. Parental knowledge of a diagnosis came from a variety of experiences: knowing their child, and prior experience, was a chief way of gauging severity and the need to consult (4.3.7). Other parents had conducted internet research to find a diagnosis (4.3.8). Two parents were healthcare professionals, and relied on their professional expertise (4.3.9) whereas some parents described not having an idea of the likely diagnosis (4.3.10).

There was some confusion over the role of antibiotics in treating LRTI. Some parents understood that antibiotics were only indicated for bacterial infections (4.3.11), while others were unclear of the difference between bacterial and viral infections and did not know when antibiotics were indicated (4.3.12), as seen in the current literature.[12]

## DISCUSSION
The study identified four themes that related to parents' perspectives, attitudes and concerns regarding the management of LTRI in children in primary care.

## Main findings
Overall, this study provides insight into the views of parents of children who have consulted with their GP for suspected LRTI. It presents up-to-date evidence of the most significant concerns for parents, primarily sudden deterioration, breathing difficulties or the child's young age. While most believed that antibiotics were effective for the symptoms of LRTI, many parents did not present with a certain expectation for antibiotics, and yet many felt frustration if the GP did not agree with their perception of the need to actively treat their child. There was a tension between parents wanting symptomatic relief for their individual child, the doctor's expertise and their inclination to prescribe based on their understanding of need in the population in general.

When asked, many parents did not recall being given specific advice on the infection duration. However, interviews revealed that parents did have some understanding about likely duration of an LRTI as shown through the 'tipping points' discussed previously. NICE guidance[5] advises patients should be advised about the natural history of RTIs including usual total length. Interviews suggest this is important for GPs to emphasise so parents can be reassured about when to present.

## Comparison with existing literature
Clinicians prescribe for medical indication and other reasons such as perceived pressure from parents,[1 14] but this pressure is not always evident. Inappropriate antibiotic prescriptions may result in problems such as antimicrobial resistance[3 12] as well as the medicalisation of a self-limiting illness.[14 16] While this study showed many parents wanted antibiotics, some indicated that they were happy to withhold antibiotic treatment and continue with non-medical self-management strategies and over-the-counter medicines in a context of reassurance, and when given a sense of illness legitimation. Parental satisfaction was likely to be highest when the GP was perceived to be thorough, that they were listened to and had their concerns taken seriously. Where appropriate, having the medical history of the child or family taken into account also helped with parental satisfaction.[14]

This study shows that parents often waited and consulted when they had reached a 'tipping point' at which they felt the severity and duration of the child's illness was such that some action in the form of prescribing antibiotics was warranted. This resonates more generally with research into urinary tract infections[17] in which respondents described an initial reluctance to consult. However, a visit was eventually prompted by symptom severity, often in conjunction with duration. This idea is echoed in research assessing clinicians' prescribing practice[14]: in the uncertainty of whether this child needs antibiotics, clinicians have their own tipping point. This could be influenced by factors such as whether the parent has previously consulted, proximity to the weekend or concerns that the parent would not recognise a clinical deterioration. Corroborating evidence with this study lends credibility to our research, adding to this important area in which we need to build a rich understanding. Previous research has identified a need to share knowledge with patients

to ensure clear understanding of antibiotics and their appropriate use.[12 15] However, this present study also showed that even if parents had a good understanding of LRTI and its causes, the perceived vulnerability of their child could influence them to present sooner than they may have done had the child been older and more able to communicate how they were feeling.

### Implications for clinical practice

Overall, most parents were satisfied with their child's GP consultation. Higher satisfaction was evident in parent narratives when parents believed their concerns about their child's health had been taken seriously, from a thorough examination to having more time spent with them. However, the tension between the parent and GP remains evident.[12] Increased GP sensitivity to the wider concerns of parents and the need to employ communication strategies that encourage parental participation and validation of their concerns could ease the path towards improved parent/patient and GP antibiotic discussions and ultimately towards more prudent antibiotic prescribing and use.

The interviews showed that while some parents did present to their GP with the expectation of receiving an antibiotic prescription, many parents did not: other aims for the consultation included illness legitimisation and advice for symptomatic relief using home management strategies. GPs need to be aware that parents are often willing to accept a treatment other than antibiotics, and often would rather have an alternative non-antibiotic approach. Providing structured information and exploration of concerns can help negotiate lower antibiotic use[18] as it does among adults.[19] Parental willingness to consider alternative management strategies often linked with the knowledge parents had about the causes of LRTI and appropriate treatment. Across the interviews, parents displayed varying levels of understanding and beliefs about causes, duration and treatment of LRTI and when they should present. Clarifying patient information, and helping to enhance general understanding, could help reduce unnecessary GP consultations and smooth the interactional path when parents do consult.

### Strengths and limitations

The sample of participants is not representative of the wider population and just sensitises us to key experiences and understandings of a sample of parents. Participants may have had stronger opinions or other reasons for being interviewed than 'typical' or non-participant parents. Only two fathers were included in interviews (compared with 23 mothers), so fathers' views were not fully captured in this study. Interviews occurred at varied times in relation to the infection/index consultation for the consulting 'child', which means that some parents had differing lengths of recall, although for most it was only a matter of weeks. Finally, interviews provide insight into perspectives on events not a direct window to the event themselves (such as the GP appointment).

However, semistructured interviews were the optimal data collection method given the aims of this research project.

### CONCLUSION

This study aimed to explore factors important to parents in the management of LTRI in children in primary care. Parental views were varied, but clearly indicated that symptomatic relief is a key consideration for their decision making regarding when to consult and management preferences. Parents do not necessarily expect an antibiotic prescription and a satisfactory consultation can be achieved by being perceived as thorough and in legitimating parents' reason for consulting. There is an ongoing need for GPs to explore concerns and expectations carefully, and to tailor advice, information and reassurance for parents—particularly addressing the natural history, worrying symptoms and the likely impact of antibiotics on symptom severity, duration and complications.

**Acknowledgements**  The interviewees were acknowledged for their invaluable contributions on this important topic. NIHR Fellowship funding was awarded to GML during the conduct of this research.

**Contributors**  GML was responsible for all aspects of the study including design, data analysis and interpretation, and manuscript preparation. AH and CvH collected, analysed and interpreted the data, and prepared the manuscript led by GML. PL is the principal investigator of a larger affiliated study, Antibiotics for lower Respiratory Tract Infection in Children presenting in Primary Care (ARTIC PC). PL and TV were involved in the generation of the study idea and design, and manuscript comments and revisions.

**Funding**  This project was funded by the National Institute for Health Research Health Technology Assessment Programme (project number 13/34/64).

**Competing interests**  None declared.

**Ethics approval**  NISCHR RES.

**Provenance and peer review**  Not commissioned; externally peer reviewed.

**Data sharing statement**  No further data are available as this was not part of the ethics approval.

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
