## [Reviewer comments · BMJ Open]

ARTICLE DETAILS

TITLE (PROVISIONAL)	A qualitative interview study of parents' perspectives, concerns and experiences of the management of lower respiratory tract infections in children in primary care.
AUTHORS	Halls, Amy; van't Hoff, Catherine; Little, Paul; Verheij, Theo; Leydon, Gerry

VERSION 1 - REVIEW

REVIEWER	Clodna McNulty Primary Care Unit, Public Health England, UK Dr McNulty writes and leads National PHE Antibiotic Guidance for Primary Care and the TARGET antibiotic resources, which include a webinar on optimising antibiotic use in children, and a webinar on use of delayed antibiotics with Paul Little, one of the authors of this paper. However, I do not consider these to influence my constructive review.
REVIEW RETURNED	23-Jan-2017

GENERAL COMMENTS	In response to the 'No' answers above: 2. Methods are lacking4. Exact recruitment method not sufficient5. Not mentioned in main text but I assume elsewhere8. I think further search needed10. Some clarification required12. I think possible selection biases - further work is needed13. There is a qualitative publication checklist which it would be good for the authors to go through15. Some lack of clarity or misuse of words in a few places Further comments from reviewer: I think the subject area for this paper is very important but the work is let down by lack of clarity of methods and if this reduces validity of results. If the points below are addressed it would be an excellent publication. 1) P4, line 16 - '...and with antibiotics' - suggest change to 'and use of antibiotics leads to greater risk of development of antibiotic resistant bacteria'2) P4, line 18 - under 4 years3) P4, line 32+33 - there is a publication in 2016 covering qualitative work with parents which has not been referenced. This is by the group in Bristol including Cabral and Hay, I think the first author is Heywood.4) P4, line 46 -
--

- a) How were individual practices chosen from the PCRN?
b) How many practices?
c) How many parents of the patients identified by searches were approached and how, how many consented and then interviewed?
5) P4, line 51 - Was it just parents or could it have been carers?
6) P5, line 4 - Where were they interviewed, by whom, when in relation to the consultation, how long did interviews take, were interviewers clinicians, were interviewers conversant with management of children in primary care?
7) P5, line 20 - Were a proportion of transcripts analysed by another researcher? Were themes discussed by the research team before full analysis?
8) P5, line 38 -
a) How many were approached from how many practices to reach 23 participants?
b) Were practices rural or urban or mix, was there any stratification by antibiotic use?
c) What was the ethnic make up?
9) P5, table 1 - It looks like median age of mothers was older than median age of all mothers or at least the group contains older mothers - if so this should be on limitations; or discussed.
10) P9, line 6 - How many children had underlying health conditions and what were they? This should be in table 1. If this makes up more than half I think the group would be considered atypical.
11) P9, line 25+26 - Repetition of line 13, 14 - delete
12) P12, line 36 - This comparison
13) P12, line 43 - Did you collect information on parents educational or social status, as this effects antibiotic use and knowledge? Did the sampling strategy aim for a range?
14) P13, line 3 - What do you mean by 'tacit'?
15) P13, line 29 - This relates to P12, line 43 - it looks like some parents were health professionals, how many? Were they a typical group, if not you need to be aware of any biases.
16) P14, line 6 - Do you mean 'disposal' or disposition, or something else? And does this refer to the doctors or the patient?
17) P14, line 23 - comparison with existing literature - need to highlight and compare with recent Bristol work published in 2016 - see attached files
18) P14, line 47 - Ref 15 - It would be better to refer to a study of UTI in children, or RTI in adults. UTI in adults seems an unusual comparator - some of the references already cited may be better (McNulty et al, Francis et al)
19) P14, line 53 - This study - do you mean ref 16 or this present study - clarify.
20) P14, line 23-58 - I am surprised there isn't some other work in parents with which to compare, or compare with copious work around RTIs in adults and confirm if very similar or not. It would be good in the implication to say whether parents can be managed in a similar way to adults with RTI or differently.
21) P16, line 13 - Do you mean any prescription - as this could be paracetamol or an antibiotic prescription.
22) P16, line 21-26 - unnecessary
23) P19-20 - Questions: some of the data collected isn't discussed in this paper. Either delete questions, or better still, explain that part of a longer interview.
24) P2, line 14 - Please say how many practices, that PCRN practices and where in the UK - one area or more than one area.
25) P2, line 18 - Please say all had consulted timing in relationship to consultation
26) Please say if consultations at GP practice had to be face-to-face,

	or could be telephone, or could be with any health professional - did this make a difference to satisfaction? The reviewer also uploaded a file in addition to these comments. Please contact the publisher for full details.
--	---

REVIEWER	Jennifer Ingram University of Bristol, UK
REVIEW RETURNED	13-Feb-2017

GENERAL COMMENTS	This is an interesting and well written paper. The main omission is that the literature review is rather out of date (very little since 2013) and fails to include a series of qualitative papers published by researchers in Bristol between 2013 and 2016 which should be included, as their work covers very similar topic areas. These papers should be included in the background and your findings compared with theirs in the discussion. In light of these papers you will need to reflect on what this study adds to the existing published research. There is no mention of ethical approval for this study. I assume this is an omission. Thorough proof reading of the final draft is encouraged as there are some grammatical errors. Detailed comments. METHODS page 5: Where did the 20 face-to-face interviews take place? FINDINGS page 9, line 15: Do you mean table 4 not 5? Line 16: Do you need any more detail about Calpol and Nurofen for an international audience? Line 20: Probably retain the past tense: ..'unless they felt it was strongly needed...' Lines 25-28: these are repeated from above. Page 12, line 25: Move 'often: 'This often seemed to be closely related...' DISCUSSION page 14, line 7: Remove split infinitive:... 'expertise and their disposal not to prescribe..' Line 23: Compare your findings with the up to date literature which has similar findings. Line 51: Remove split infinitive: ..'research has identified a need to share knowledge with patients...' STRENGTHS AND LIMITATIONS Page 15, line 52: ..'parents had differing lengths of recall.' CONCLUSION Page 16, line 18: Remove split infinitive:... 'need for GPs to explore concerns and expectations carefully,...' Line 22: See Cabral et al in Annals of Family Medicine 2015 paper (The influence of clinical communication on parents' antibiotic expectations for children with Respiratory Tract Infections) for report of a video study of GP consultations.
---

REVIEWER	Lourdes Cantarero-Arévalo, Assistant professor Faculty of Health and Medical Sciences University of Copenhagen
REVIEW RETURNED	17-Feb-2017

GENERAL COMMENTS	This is a correct and clear scientific article aiming at exploring parental perspectives and concerns related to the management of lower respiratory track infections and use of antibiotics. The article well structured and includes the main components of a qualitative-based piece of research. However there are major concerns mainly with the "newness" of the research question and therefore its findings. This article does not really add anything new in the field of parental perceptions and concerns of management of LRTIs and use of AB. Only the UK there is a wealth of research conducted in the field using qualitative methods as well as survey-based and several systematic reviews are available summarizing the collective findings. It is therefore my suggestion not to accept this Scientific article for publication. My suggestion to the authors in case they want to pursue the publication of their work is the following:  1. complement the interviews among parents with, when possible observations of encounters between GPs parents and children, and/or 2. focus groups with parents and their GPs to better understand the interactions between the two actors 3. Make use of theory to inform the interview guide and therefore provide more depth to the data
--

VERSION 1 – AUTHOR RESPONSE

Reviewer 1

I think the subject area for this paper is very important but the work is let down by lack of clarity of methods and if this reduces validity of results. If the points below are addressed it would be an excellent publication.

Comment:

“1) P4, line 16 - '...and with antibiotics' - suggest change to 'and use of antibiotics leads to greater risk of development of antibiotic resistant bacteria’”

Response:

Thank you, we have made this change (page 4).

Comment:

“2) P4, line 18 - under 4 years”

Response:

Thank you, this has been changed (page 4).

Comment:

“3) P4, line 32+33 - there is a publication in 2016 covering qualitative work with parents which has not been referenced. This is by the group in Bristol including Cabral and Hay, I think the first author is Heywood.”

Response:

Thank you, this has now been referenced (page 4). "Previous work has assessed parents' understanding of the implications and indications for an antibiotic prescription(12) , with parents believing that antibiotics were for more severe illnesses, supported by GPs explanations. This study provides further insight into the tension between GPs and parents in their decision making, and explores key drivers of parental expectation and beliefs about antibiotics."

Comment:

"4) P4, line 46 -

- a) How were individual practices chosen from the PCRN?
- b) How many practices?
- c) How many parents of the patients identified by searches were approached and how, how many consented and then interviewed?"

Response:

Thank you for these questions. We have addressed them as follows:

- a) Practices were chosen by the PCRN if they were research-active and willing to participate in research of this nature.
- b) Six practices participated and we have added this to the manuscript (page 4) "Six research-active practices in the South West of England were identified through the Primary Care Research Network and a poster presentation by CvH at the Wessex Research Sites Initiative Conference."
- c) A flow diagram has been added to the manuscript (Fig1, page 5) together with a table of participating practices (this is now Table 1, page 5).

Comment:

"5) P4, line 51 - Was it just parents or could it have been carers?"

Response:

Thank you for this comment. All caregivers were invited to participate, it just so happened to be only parents who participated. This is clarified on page 6: "In all interviews the mother of the child was interviewed, and fathers were present in two interviews, however anyone who fulfilled a caring role could have participated."

Comment:

"6) P5, line 4 - Where were they interviewed, by whom, when in relation to the consultation, how long did interviews take, were interviewers clinicians, were interviewers conversant with management of children in primary care?"

Response:

Thank you for this comment – we agree this information is important and should have been included in the original manuscript. We have provided more information on page 6 to strengthen this section.

"Two interviewers (CvH and AH) conducted face to face interviews in the participant's home (n=20) and telephone interviews (n=3), each lasting approximately 30 to 60 minutes, with an average duration of 42 minutes. Both interviewers were aware of the NICE guidance regarding antibiotic prescribing for RTIs. All interviews were audio-recorded and transcribed verbatim. Qualitative interviews provided the best method for gathering insights into parents' views about their children's

experiences of RTI and its management in primary care. As part of a longer interview, the semi-structured interview guide (Appendix 1) included key topic areas whilst also giving flexibility to explore new areas if they arose.”

Comment:

“7) P5, line 20 - Were a proportion of transcripts analysed by another researcher? Were themes discussed by the research team before full analysis?”

Response:

Thank you for this comment, we have elaborated on the analysis process on page 6: “Inductive thematic analysis(13) was conducted on all transcripts to gain an understanding of the perspectives, attitudes and concerns of parents regarding the management of LRTI in children in primary care. CvH and AH achieved familiarisation through repeated reading (and listening) of the transcripts. Patterns and themes in the data were identified by CvH and AH and labelled with codes: these code labels referred to the operationalisation of the theme content. A label and full description were provided for each theme. These codes and definitions were refined during a continuous process led by GL, CvH and AH. This involved themes being linked, grouped, moved, re-labelled, added and removed as appropriate in order to produce a set of themes, subthemes, and a coding manual, reviewed and confirmed by the full research team agreed. The coding manual thoroughly explained all of the data.”

Comment:

“8) P5, line 38 -

- a) How many were approached from how many practices to reach 23 participants?
- b) Were practices rural or urban or mix, was there any stratification by antibiotic use?
- c) What was the ethnic make up?”

Response:

Thank you, again this should have been included in the original manuscript. We have provided more information. The addition of Fig 1 (page 5) shows the process of recruitment. Table 1 (practice characteristics and recruitment, also page 5) includes the deprivation index and we have added ethnicity to the participant characteristics table (now numbered as Table 2, page 7).

Comment:

“9) P5, table 1 - It looks like median age of mothers was older than median age of all mothers or at least the group contains older mothers - if so this should be on limitations; or discussed.”

Response:

Thank you for this comment. We would appreciate it if the reviewer could expand upon this as we are uncertain how to respond. The median age is correct, there are older mothers in the sample but there is a range of ages, especially when considering the age of the children. We do not think that the age range of mothers participating in this study is a limitation.

Comment:

“10) P9, line 6 - How many children had underlying health conditions and what were they? This should be in table 1. If this makes up more than half I think the group would be considered atypical.”

Response:

Thank you for this comment, we have added a column to what is now Table 2. Nine out of 23 children had a suspected or diagnosed underlying health condition. This is on page 7.

Comment:

“11) P9, line 25+26 - Repetition of line 13, 14 – delete”

Response:

We have deleted the lines as suggested.

Comment:

“12) P12, line 36 - This comparison”

Response:

We are unsure what the reviewer means here. Please could we have further advice?

Comment:

“13) P12, line 43 - Did you collect information on parents' educational or social status, as this effects antibiotic use and knowledge? Did the sampling strategy aim for a range?”

Response:

Thank you, we have provided more information on page 5: “The recruitment process is shown in Figure 1. A purposive sample was originally intended: however, due to initially slow recruitment, a convenience sampling strategy was used. Whilst this sampling approach can be limiting we recruited a good range of mothers in terms of age and occupation.”

Comment:

“14) P13, line 3 - What do you mean by 'tacit'?”

Response:

We have edited this sentence on page 14 and hope it is now clearer: “Parents had a degree of prior knowledge regarding a suggested diagnosis, when to present, and if they felt antibiotics were indicated.”

Comment:

“15) P13, line 29 - This relates to P12, line 43 - it looks like some parents were health professionals, how many? Were they a typical group, if not you need to be aware of any biases.”

Response:

Thank you for this comment. We have clarified on page 14 that just two parents worked in health care: “Two parents were health care professionals, and relied on their professional expertise (4.3.9)

whereas some parents described not having an idea of the likely diagnosis (4.3.10).”

Comment:

“16) P14, line 6 - Do you mean 'disposal' or disposition, or something else? And does this refer to the doctors or the patient?”

Response:

Thank you for highlighting that this is unclear. We have edited this sentence on page 15. It now reads as follows: “There was a tension between parents wanting symptomatic relief for their individual child, the doctor’s expertise and their inclination to prescribe based on their understanding of need in the population in general.”

Comment:

“17) P14, line 23 - comparison with existing literature - need to highlight and compare with recent Bristol work published in 2016 - see attached files”

Response:

Thank you for attaching these papers. They are references 12 and 14 and have been added to the ‘Comparison with existing literature’ section on pages 15 and 16.

Comment:

“18) P14, line 47 - Ref 15 - It would be better to refer to a study of UTI in children, or RTI in adults. UTI in adults seems an unusual comparator - some of the references already cited may be better (McNulty et al, Francis et al)”

Response:

Thank you for this comment. We have edited the text on page 16. Due to adding references in the resubmission process, ref 15 is now ref 17. “This resonates more generally with research into urinary tract infections(17) in which respondents described an initial reluctance to consult. However, a visit was eventually prompted by symptom severity, often in conjunction with duration. This idea is echoed in a research assessing clinicians’ prescribing practice(14): in the uncertainty of whether this child needs antibiotics, clinicians have their own tipping point. This could be influenced by factors such as whether the parent has previously consulted, proximity to the weekend, or concerns that the parent would not recognise a clinical deterioration.”

We are comparing a ‘tipping point’ for adults in terms of consulting for themselves (UTIs) and for their children (RTIs). We believe this comparison to be valid as it is the symptom severity which has prompted adults to consult, either on their own behalf or that of their child’s.

Comment:

“19) P14, line 53 - This study - do you mean ref 16 or this present study - clarify.”

Response:

Thank you for this observation. We have edited the text on page 16 to read as follows, which should

now be clearer:

“However, this present study also showed that even if parents had a good understanding of LRTI and its causes, the perceived vulnerability of their child could influence them to present sooner than they may have done had the child been older and more able to communicate how they were feeling.”

Comment:

“20) P14, line 23-58 - I am surprised there isn't some other work in parents with which to compare, or compare with copious work around RTIs in adults and confirm if very similar or not. It would be good in the implication to say whether parents can be managed in a similar way to adults with RTI or differently.”

Response:

Thank you for this comment. We have edited the paragraph over pages 16 and 17 to read as follows which hopefully is clearer: “The interviews showed that whilst some parents did present to their GP with the expectation of receiving an antibiotic prescription, many parents did not: other aims for the consultation included illness legitimisation and advice for symptomatic relief using home management strategies. GPs need to be aware that parents are often willing to accept a treatment other than antibiotics, and often would rather have an alternative non-antibiotic approach. Providing structured information and exploration of concerns can help negotiate lower antibiotic use(18)) as it does among adults(19). Parental willingness to consider alternative management strategies was often linked with the knowledge parents had about the causes of LRTI and appropriate treatment.”

Comment:

“21) P16, line 13 - Do you mean any prescription - as this could be paracetamol or an antibiotic prescription.”

Response:

Thank you for this observation. It has been edited on page 17 to now read ‘antibiotic prescription’.

Comment:

“22) P16, line 21-26 – unnecessary”

Response:

These sentences have been removed.

Comment:

“23) P19-20 - Questions: some of the data collected isn't discussed in this paper. Either delete questions, or better still, explain that part of a longer interview.”

Response:

Thank you for this observation. Page 6 has now been edited as follows and we hope this is clearer: “The interview guide (Appendix 1) included key topic areas, not all of which are covered in this paper. Interviews explored parent views on whether they would be willing to be randomised in a future study.

Its semi-structured design gave flexibility to explore new areas if they arose.”

Comment:

“24) P2, line 14 - Please say how many practices, that PCRN practices and where in the UK - one area or more than one area.”

Response:

Thank you for this comment, the section Participants and procedure on page 4 has been edited as follows, which we hope provides more information:

“Six research-active practices in south west England were identified through the Primary Care Research Network and a poster presentation at the Wessex Research Sites Initiative Conference.”

Comment:

“25) P2, line 18 - Please say all had consulted timing in relationship to consultation”

Thank you for this comment. The section Strengths and limitations has been edited on page 17 to reflect this and reads as follows: “Interviews occurred at varied times in relation to the infection/index consultation for the consulting ‘child’, which means that some parents had differing lengths of recall.”

Comment:

“26) Please say if consultations at GP practice had to be face-to-face, or could be telephone, or could be with any health professional - did this make a difference to satisfaction?”

Response:

Thank you for this comment. This has been addressed in Participants and procedure on page 5. We did not systematically enquire whether participants consulted face-to-face or by telephone. But we do know that all consultations were with a GP and, based on interview content, can infer that most participants had been recruited following face to face consultation.

Reviewer 2

Reviewer: 2

Reviewer Name: Jennifer Ingram

Institution and Country: University of Bristol, UK

Please state any competing interests or state ‘None declared’: None declared

Please leave your comments for the authors below

This is an interesting and well written paper.

The main omission is that the literature review is rather out of date (very little since 2013) and fails to include a series of qualitative papers published by researchers in Bristol between 2013 and 2016 which should be included, as their work covers very similar topic areas. These papers should be included in the background and your findings compared with theirs in the discussion.

In light of these papers you will need to reflect on what this study adds to the existing published research.

There is no mention of ethical approval for this study. I assume this is an omission.

Thorough proof reading of the final draft is encouraged as there are some grammatical errors.

Detailed comments.

METHODS

Comment:

“page 5: Where did the 20 face-to-face interviews take place?”

Response:

Thank you for this comment. We have edited page 6 to read as follows: “Two interviewers (CvH and AH) conducted face to face interviews in the participant’s home (n=20) and telephone interviews (n=3), each lasting approximately 30 to 60 minutes, with an average duration of 42 minutes.”

FINDINGS

Comment:

“page 9, line 15: Do you mean table 4 not 5?”

Response:

Thank you for this observation. It was originally incorrect, and should have been Table 4. We have since added a new table (Table 1) so this table on what is now page 10 is correctly labelled as Table 5.

Comment:

“Line 16: Do you need any more detail about Calpol and Nurofen for an international audience?”

Response:

A footnote has been added to page 10 and reads as follows: “Analgesics widely available in the United Kingdom: Calpol contains paracetamol and Nurofen contains ibuprofen and codeine phosphate.” We hope this provides sufficient information for an international audience.

Comment:

“Line 20: Probably retain the past tense: ..’unless they felt it was strongly needed...”

Response:

Thank you for this comment. The sentence has been edited on page 10 and now reads as follows: “Perhaps unsurprisingly many parents reported reluctance to use medications unless they felt it was strongly needed or recommended by a doctor (1.5.3).”

Comment:

“Lines 25-28: these are repeated from above.”

Response:

Thank you for this observation. These lines have been deleted.

Comment:

“Page 12, line 25: Move ‘often: ‘This often seemed to be closely related...”

Response:

Thank you, this has been moved as suggested (page 13).

DISCUSSION

Comment:

“page 14, line 7: Remove split infinitive:...’expertise and their disposal not to prescribe..”

Response:

Thank you. This has been edited as suggested.

Comment:

“Line 23: Compare your findings with the up to date literature which has similar findings.”

Response:

Thank you for this observation, which echoes that of Reviewer 1. We have added in additional, more recent, publications.

Comment:

“Line 51: Remove split infinitive: ..’research has identified a need to share knowledge with patients...”

Response:

Thank you, this has been corrected.

STRENGTHS AND LIMITATIONS

Comment:

“Page 15, line 52: ..’parents had differing lengths of recall.”

Response:

Thank you, this sentence has been edited on page 17 and reads as follows: “Interviews occurred at varied times in relation to the infection/index consultation for the consulting ‘child’, which means that some parents had differing lengths of recall, although for most it was only a matter of weeks.”

CONCLUSION

Comment:

“Page 16, line 18: Remove split infinitive:...'need for GPs to explore concerns and expectations carefully,...”

Response:

Thank you, this has been edited.

Comment:

“Line 22: See Cabral et al in Annals of Family Medicine 2015 paper (The influence of clinical communication on parents' antibiotic expectations for children with Respiratory Tract Infections) for report of a video study of GP consultations.”

Response:

Thank you for this recommendation. It has been added and is reference 12.

Reviewer: 3

Reviewer Name: Lourdes Cantarero-Arévalo, Assistant professor

Institution and Country: Faculty of Health and Medical Sciences, University of Copenhagen

Please state any competing interests or state 'None declared': None declared

Please leave your comments for the authors below

This is a correct and clear scientific article aiming at exploring parental perspectives and concerns related to the management of lower respiratory tract infections and use of antibiotics. The article is well structured and includes the main components of a qualitative-based piece of research. However there are major concerns mainly with the "newness" of the research question and therefore its findings. This article does not really add anything new in the field of parental perceptions and concerns of management of LRTIs and use of AB. Only in the UK there is a wealth of research conducted in the field using qualitative methods as well as survey-based and several systematic reviews are available summarizing the collective findings. It is therefore my suggestion not to accept this Scientific article for publication. My suggestion to the authors in case they want to pursue the publication of their work is the following:

1. complement the interviews among parents with, when possible observations of encounters between GPs parents and children, and/or
2. focus groups with parents and their GPs to better understand the interactions between the two actors
3. Make use of theory to inform the interview guide and therefore provide more depth to the data

Thank you for your comments, we are pleased you found it to be well-structured and clear. We have added in more recent publications, following suggestions from Reviewers 1 and 2 and our research corroborates key findings from these papers. This is an important and topical area, in which we need to build a rich understanding. Our findings showed that the perceived vulnerability of young children could influence parents to present sooner than they may have done had the child been older, or better able to communicate, even if the parents had a good understanding of LRTI and its causes. It also showed there is a need to clarify patient information, which could help reduce unnecessary GP consultations. We believe this study complements recent research conducted in this area, as well as adding to the developing knowledge base. Thank you for the suggestions as to how we could develop this research further, and these will be duly considered.